# Time-Tested Strategies in Modern Context: A Bibliometric Study of Chemotherapy’s Continued Importance in Colorectal Cancer Treatment

**DOI:** 10.3390/cancers15184520

**Published:** 2023-09-12

**Authors:** Helena Clar-Marmaneu, Alba Estela García-Fernández, Francisco Javier García-Fernández

**Affiliations:** 1Facultad de Farmacia, Universitat de València, 46100 Burjassot, Spain; 2Informática Médico Farmaceútica, 46021 Valencia, Spain; 3Bioquímica Clínica, Hospital Vall d’Hebron, 08023 Barcelona, Spain; 4Servicio de Análisis Clínicos, Hospital General de Tomelloso, 13700 Tomelloso, Spain

**Keywords:** colorectal cancer, chemotherapy, 5-fluorouracil, capecitabine, irinotecan, oxaliplatin, trifluridine, tipiracil

## Abstract

**Simple Summary:**

In the treatment of colorectal cancer, classic chemotherapy drugs such as 5-fluorouracil, capecitabine, irinotecan, oxaliplatin, trifluridine, and tipiracil have played a crucial role. Through the analysis of the top 100 most influential articles, we examined the evolution in research and current relevance, confirming the continued significance of this group of drugs despite the emergence of new treatments. The research reveals global collaboration among institutions, countries (primarily the United States, China, and Europe), and researchers, with three main themes driving the study: pharmacogenetics, new pharmaceutical formulations, and the use of adjuvants.

**Abstract:**

In the landscape of colorectal cancer treatment, classical chemotherapeutic agents such as 5-fluorouracil, capecitabine, irinotecan, oxaliplatin, trifluridine, and tipiracil have historically played a pivotal role. This study presents a comprehensive bibliometric analysis of the top 100 most influential articles focusing on these classic chemotherapy drugs in the management of colorectal cancer. With this, we shed light on their current importance, despite the emergence of new therapeutic targets and treatments in the field of oncology. Systematically evaluating research outputs, this analysis reveals a prevalence of co-authorship among institutions, countries (led by the United States, China, and Europe), and researchers highlighting the global and collaborative nature of efforts in research, utilization, and development of these drugs. Three thematic axes lead the research: pharmacogenetics, the development of new pharmaceutical forms, and the use of adjuvants. This research serves as a foundation for future endeavors, aiding researchers, clinicians, and policymakers in making informed decisions about the direction of research and development in the dynamic field of colorectal cancer therapy.

## 1. Introduction

Colorectal cancer, comprising cancers of the colon and rectum, has demonstrated the highest incidence and mortality rates among all cancer types in the United States, as per the NIH data presented in its Cancer Facts and Figures 2020 document [1]. In 2020, 147,950 new cases were diagnosed, and 53,200 patients succumbed to the disease. Drug therapy represents one of the primary modalities available for the treatment of colorectal cancer. Traditional or standard chemotherapy [2] utilizing cytotoxic drugs (5-fluorouracil, capecitabine, irinotecan, oxaliplatin, trifluridine, and tipiracil) [3,4] to eliminate tumor cells continues to be an option despite the emergence of new pharmaceutical technologies. These drugs can be administered alone or in combination. The significant scientific interest in discovering new drugs against colorectal cancer or investigating existing drugs has resulted in a voluminous number of publications on these substances. Bibliometrics employs mathematical and statistical methods to quantitatively analyze scientific activity, making it possible to compare, measure, and objectify scientific activity. As a result, it provides valuable insights into the behavior of science and scientists [5].

In our study, we will leverage the two main and distinctive functions of bibliometrics. Firstly, we will utilize the mathematical and statistical methods provided by bibliometrics to select the most relevant articles. Given the existence of thousands of publications on the use of cytotoxic drugs in the treatment of colorectal cancer, it would be impractical for any researcher to review them all. Therefore, through bibliometrics, we will identify the 100 most important articles that will be crucial for our analysis, allowing us to evaluate whether chemotherapy still has a role in the investigation of treatments for this disease, as well as identifying current and potential lines of research for the future.

Secondly, we will use bibliometrics as an analytical tool to identify the key players who have led the research in this field. We will explore the existing collaborations among them, which will allow us to gain a better understanding of the research networks and the interaction among key researchers. This approach will provide us with a more comprehensive view of the scientific landscape regarding the use of cytotoxic drugs in the treatment of colorectal cancer.

We can identify the emerging themes that will guide future research, the authors or institutions that have been instrumental in exploring the most effective treatments, and the geographic distribution of countries at the forefront of research. Furthermore, we can ascertain the scientific journals with the broadest dissemination and acceptance among the research community. Citations serve as the foundation of bibliometric analysis and are among the best indicators of the quality of a study [6] and an objective criterion for selecting works to conduct research.

## 2. Materials and Methods

On 20 April 2023, we performed the last bibliographic search in Web of Science (WOS, Clarivate Analytics, Philadelphia, PA, USA) for all articles containing the terms “5-fluorouracil”, “5-FU”, “capecitabine”, “xeloda”, “irinotecan”, “camptosar”, “oxaliplatin” eloxatin”, “trifluridine and tipiracil”, in order to collect the different names of the main chemotherapy treatments used in colorectal cancer, both conventional drugs and new therapeutic targets. To ensure comprehensive drug selection, we used the cancer.gov website as a reference and extracted both the trade name and the FDA-accepted active ingredient name for each medication belonging to the American Cancer Society, and in the bibliography provided by this society [7].

The databases used to obtain the papers were: SCI-EXPANDED, Social Science Citation Index (SSCI), Arts and Humanities Citation Index (A & HCI), Conference Proceedings Citation Index-Science (CPCI-S), Conference Proceedings Citation Index-Social Science and Humanities (CPCI-SSH), Emerging Sources Citation Index (ESCI), Current Chemical Reactions-Expanded (CCR-EXPANDED) and Index Chemicus (IC). No filters were applied by time, article language, authors, participating institutions, topics or article funding.

After conducting preliminary searches to contextualize the study and in accordance with the total number of researchers, it was decided that the final analysis would be limited to entries that included the name of the drug as part of the article title, excluding those in which it appeared only in the abstract.

A total of 42,276 articles were identified. To focus exclusively on original research articles, we excluded Meeting Abstracts, Review Articles, Proceedings Papers, Editorial Materials, Early Access, Corrections, News Items, Book Chapters, Retracted Publications, and Retractions. This left us with a total of 11,024 articles, consisting of 10,610 Articles and 414 Letters. On the day of the search, these articles collectively had accumulated 172,123 citations, with an average of 15.61 citations per article.

To narrow our focus to articles specifically related to colorectal cancer, we implemented a secondary filter using keywords such as “colorectal”, “colon”, and “rectal”. Subsequently, we organized the articles based on their citation counts and selected the top 100 most cited articles (T100). Within this subset of articles, we conducted an analysis of various variables, including subject matter, authorship, title, citation count, source, author identification, the specific institution or country where each T100 article was published, citation density (citations per year), and citations per record.

The data obtained from this search were exported to a Microsoft Excel spreadsheet located in Redmond, WA, USA. It is worth noting that none of the authors of this manuscript have affiliations or associations with pharmaceutical industries involved in the research or production of these drugs.

## 3. Results

On 20 April 2023, a search was conducted to gather articles for a subsequent bibliometric study focused on the chemotherapeutic treatment of colorectal cancer. Initially, a total of 42,486 articles were retrieved by entering relevant terms into the Web of Science (WOS) database. However, after refining the search to include only original publications, we selected a total of 11,131 papers, comprising 10,621 articles and 510 letters. These publications collectively amassed 178,433 citations, resulting in an average citation density of 16.03 citations per article.

Subsequently, we applied a filter to isolate articles related to colorectal cancer, which yielded a total of 2504 papers, including 2132 articles and 372 letters. The cumulative number of these papers accumulated 48,009 citations, with an average citation density of 19.17 citations per article. Among these, the top 100 most cited articles (Table 1) received a total of 12,329 citations, with an average citation density of 123.29 citations per article.

The highest-cited article, published in 2004 in Annals of Oncology, garnered a total of 778 citations. It focused on the analysis of adverse reactions to chemotherapy in 153 patients with colorectal cancer, specifically examining the use of oxaliplatin. The list of the top 100 articles was completed with an article published in 2016 in the journal Biochemical Pharmacology, which received a total of 60 citations. This study investigated the addition of a natural diterpenoid called Andrographolide to enhance the response of 5-FU in the treatment of colorectal cancer.

The year 2017 stands out as having the most influential papers, with 8 articles from the top 100 published during that year. Additionally, from 2010 to the present, 55 out of the 100 articles were published, with 22 of them appearing in the last five years (since 2017). The last decade (2013–2022) contributed 41 articles, reflecting research trends in pharmacogenetics, adjuvant substances, and the synthesis of nanoparticles and liposomes.

In total, 734 authors contributed to the T100 articles. The top 3 authors with the highest number of published articles within the T100, each having authored 4 articles, are Falcone A and Loupakis F, accumulating a total of 382 citations and a citation density of 95.5 citations per article. Ranking third is Goel A, with 343 citations across his 4 articles, resulting in an average of 85.75 citations per article.

From a geographical perspective, the United States leads in production with 25 articles, amassing a total of 3939 citations. China closely follows with 24 papers, which received 2057 citations. Spain secures the third position with 6 papers, contributing to a total of 559 citations. France boasts the highest citation density per article, achieving 2306 citations across its 12 papers, resulting in an impressive density of 192.17 citations per article (Table 2).

Funding sources reveal that the National Natural Science Foundation of China (NSFC) supported 13 of the T100 papers, while US institutions such as the National Institutes Of Health and National Cancer Institute also funded 3 T100 papers each. It is worth noting that all T100 articles were composed in English.

Among the journals featuring T100 articles, Annals of Oncology takes the lead with a total of 11 articles. Furthermore, it boasts an impressive impact factor of 32.97 for the year 2021.

## 4. Discussion

The research on cytotoxic drugs is of significant current relevance, representing the primary key finding from our analysis. When examining the temporal evolution of the most relevant articles, it becomes evident that the publication distribution from year to year underscores the contemporary and substantial nature of this topic. Specifically, 55 out of the 100 relevant articles have been published from 2010 to the present day. Within the last decade, spanning from 2013 to 2022, 41 articles have been contributed. Furthermore, in the past five years, from 2017 to the present, our ranking includes 22 papers (see Figure 1).

Moreover, the year 2017 stands out with the highest number of contributions to the T100. It is noteworthy that our analysis exclusively focused on original articles, leading us to identify a total of 2470 papers with a collective citation count of 46,189, indicative of a considerable impact. This pronounced impact is also evident in the T100 dataset (refer to Table 1), where the 100 most cited articles amass a total of 12,329 citations, yielding an average citation density surpassing 120 citations per article.

To ascertain the prevailing research trends in the realm of colorectal cancer treatment, we conducted an extensive analysis of the content within the 41 articles published in the top 100 journals (T100) over the past decade. Our examination revealed three key areas of research that are currently attracting substantial attention.

The first area of research revolves around pharmacogenetics, which delves into how genetic variations influence a patient’s response to chemotherapeutic drugs used in colorectal cancer treatment [8,9]. Researchers are investigating how individual genetic differences can impact drug efficacy and toxicity, with the aim of optimizing treatment outcomes and minimizing side effects [10,11,12].

The second area of research focuses on the utilization of adjuvant substances. These are additional agents administered alongside chemotherapeutic drugs to enhance their effects. Such substances may encompass other drugs or natural compounds that have demonstrated the ability to improve drug delivery, enhance drug efficacy, or reduce drug resistance.

Lastly, the third area of research revolves around the development of more efficient pharmaceutical forms. Researchers are exploring novel drug delivery systems designed to enhance drug bioavailability and pharmacokinetics while simultaneously reducing toxicity and side effects. This includes the creation of targeted drug delivery systems capable of specifically targeting cancer cells while sparing healthy ones from harm.

In summary, our analysis of the literature indicates that these three research areas are currently the most active and promising within the field of colorectal cancer treatment.

The articles focusing on the pharmacogenetic study within the T100 dataset are #15, #21, #31, #37, #49, #51, #54, #57, #66, #67, #68, #71, #72, #73, #75, #87, and #89.

In the realm of pharmacogenetic variability in colorectal cancer therapy, it is noteworthy that non-coding molecules play a pivotal role in regulating the expression of genes associated with chemotherapeutic drug metabolism, rather than changes in coding genes. Recent research has concentrated on the study of microRNAs, which are small single-stranded RNA fragments that regulate other genes through ribo-interference. Despite their inability to produce proteins, microRNA molecules can significantly influence the effectiveness and toxicity of chemotherapeutic drugs.

Several non-coding RNA molecules have been identified as participants in regulating chemoresistance in colon cancer cells. These include MALAT1, lncRNA-like microRNA #15, Linc00152 #21, microRNA-625-3p #51, and lncRNA KCNQ1OT1 #54 and #57, all of which enhance oxaliplatin chemoresistance by targeting the miR-34a/ATG4B pathway. Conversely, miR-19b-3p promotes colon cancer cell proliferation while also fostering oxaliplatin-based chemoresistance by targeting SMAD4 #71. Additionally, miR-136 promotes metastasis and drug resistance through the long non-coding RNA CRNDE #72, while miR-637 achieves this through autophagy #75.

However, miR-506 enhances the sensitivity of human colorectal cancer cells to oxaliplatin by suppressing MDR1/P-gp expression (#89). Genetic variability also affects another chemotherapeutic drug, 5-fluorouracil, due to certain microsatellites. In one study, it was discovered that miR 139-5p, which targets NOTCH-1, and microRNA-34a, which inhibits the enzyme lactate dehydrogenase, sensitize colon cancer cells to 5-fluorouracil. Additionally, overexpression of MicroRNA-122 re-sensitizes 5-FU-resistant colon cancer cells to 5-FU by inhibiting PKM2 in vitro and in vivo (#73).

BRAF V600E and KRAS mutations have been extensively studied and found to be significantly associated with shorter disease-free survival and overall survival in patients with microsatellite-stable tumors but not in patients with microsatellite-unstable tumors. Investigating the impact of microorganisms on variable gene expression that can influence pharmacogenetics is crucial (#31). Fusobacterium nucleatum promotes chemoresistance to 5-fluorouracil by positively regulating BIRC3 expression in colorectal cancer (#37). Furthermore, the analysis of the functional capacity of the gut microbiota using PICRUSt revealed that genes involved in amino acid metabolism, replication and translation repair, and nucleotide metabolism were better expressed in a healthy microbiota (#49).

The second trending topic involves the study of adjuvant substances that enhance the properties of chemotherapy drugs used in colorectal cancer therapy. Articles #22, #39, #43, #50, #56, #69, #79, #86, #94, #95, and #100 explore the use of adjuvant substances, such as melatonin, curcumin, or resveratrol, as well as Epigallocatechin-3-gallate, which exhibit positive effects when administered alongside cytotoxic therapy [13,14,15].

Study #22 suggests that the hormone melatonin can enhance the effectiveness of 5-fluorouracil in treating colon cancer by suppressing two signaling pathways, namely PI3K/AKT and NF-κB/Inos. Additionally, melatonin in combination with 5-fluorouracil can work together to suppress colon cancer stem cells by regulating the cellular prion protein axis (#95).

In study #43, resveratrol is found to sensitize colorectal cancer cells to 5-fluorouracil treatment by positively regulating intercellular junctions, epithelial-to-mesenchymal transition, and apoptosis. Resveratrol also enhances the effect of TNF-beta-induced cell death in 5-fluorouracil-treated colorectal cancer cells (#94).

Study #50 demonstrates that curcumin can reverse resistance to oxaliplatin in colorectal cancer cells by modulating the CXC-chemokine/NF-κB signaling pathway. Curcumin also sensitizes 5-fluorouracil-resistant MMR-deficient human colon cancer cells in high-density culture (#56).

In article #69, a combination of drugs and supramolecular therapy administered with oxaliplatin is shown to have adjuvant efficacy. Furthermore, cyclooxygenase-2 inhibitors can increase the efficacy and delivery of carboxylated pilar [6] and decrease cytotoxicity.

Two studies (#38 and #86) demonstrate that epigallocatechin-3-gallate can target cancer stem cells and enhance the sensitivity of colorectal cancer cells to 5-fluorouracil by inhibiting the GRP78/NF-kappa B/miR-155-5p/MDR1 pathway.

Work #100 reveals that andrographolide can reverse 5-fluorouracil resistance in human colorectal cancer cells by upregulating BAX expression.

Lastly, recent developments in new pharmaceutical forms aim to improve the efficacy and safety of drugs [16,17,18]. Pharmaceutical technology is being utilized to discover new delivery vehicles capable of accurately delivering drugs to their intended targets. The use of nanoparticles and liposomes has emerged as a promising approach to drug delivery, as highlighted in articles #47, #56, #92, #93, and #98.

As an example of this approach, researchers have developed Eudragit S100-coated citrus pectin nanoparticles for targeted delivery of 5-fluorouracil in the colon (#47). Another study employed enteric coating to achieve sustained and localized release of 5-fluorouracil (#56). Additionally, pH-sensitive double-layered alginate/chitosan/kappa-carrageenan microspheres were designed for controlled release of 5-fluorouracil in the colon (#92).

In addition to nanoparticles, liposomes are also under investigation for targeted drug delivery. For instance, a novel delivery system using folic acid-conjugated liposomes was developed for 5-fluorouracil in colon cancer therapy (#93). Another study explored the use of pH-sensitive ZnO/carboxymethylcellulose/chitosan bionanocomposite beads to achieve colon-specific release of 5-fluorouracil (#98).

Overall, the development of these new pharmaceutical forms represents an exciting area of research with the potential to significantly improve the effectiveness and safety of drug treatments. By leveraging advanced technology to precisely deliver drugs to their intended targets, researchers can reduce the risk of side effects and enhance the efficacy of treatments for various diseases, including cancer.

Regarding the analysis of authors who participated in the T100 articles, it is evident that strong co-authorship is prevalent among them. Collaborative efforts involving multiple professionals in research and article publication are regarded as a quality parameter, providing a broader perspective, potential multidisciplinarity, and access to a wider network of readers. Over 700 authors contributed to the T100. Co-authorship serves as an indicator of collaboration and teamwork among researchers, measuring collaboration between institutions and countries in scientific output [19].

Furthermore, co-authorship is significant for assessing the quality and impact of research, as collaboration among authors from different institutions and disciplines can bring diverse perspectives and expertise to a research project. Co-authorship also aids in evaluating researchers’ productivity and the influence of their work within the scientific community (see Figure 2 and Figure 3).

The bibliometric analysis of the authors reveals that the Lotka law of productivity does not apply in this case, as the production of articles is distributed quite evenly among many authors. This law typically suggests that a small group of authors produces the majority of relevant publications and, consequently, accumulates the majority of citations. While three authors have authored 4 articles each, making them the most prolific, they have received just over 300 citations collectively. However, it is important to note that this citation count might not be considered high when considering the high citation density per article [19].

In terms of research leadership, both the United States and China are at the forefront of this field, with nearly equal numbers of articles (25 versus 24, respectively). However, it is noteworthy that articles in which the United States participated received nearly twice as many citations. It is worth mentioning that China entered the research arena later, with its first article published in the T100 in 2006. Additionally, four European countries also contribute significantly to this field (see Table 2 and Figure 4).

These results underscore the exponential growth in research within China. According to data from the Chinese Ministry of Science and Technology, the Chinese government has substantially increased its investment in research over the past decade. In 2019, the total investment in research and development (R & D) in China reached CNY 2.44 trillion (approximately USD 371.7 billion), marking a 12.5% increase compared to the previous year. Moreover, investment in R & D has steadily risen in recent years, growing from 1.42% of GDP in 2009 to 2.23% in 2019 [20].

It is noteworthy that English serves as the essential tool for scientific communication, even in research related to the use of colorectal chemotherapy. All articles in the T100 are written in English [21]. This is understandable because, for an article to be considered relevant, it must reach the widest possible audience, and there is no language more universal than English. All articles are published in high-impact journals, with the journal Annals of Oncology being the most prolific in the T100. It is a specific journal focused on cancer research and had an impact factor of 32.97 in 2021 (see Table 3).

Additionally, the primary institutions that have funded these papers underscore the significance of this topic. Public institutions from two of the world’s most influential countries, the National Natural Science Foundation of China (NSFC) and American institutions such as the National Institutes of Health and National Cancer Institute, have provided funding various of the T100 papers. The NSFC has funded 13 papers, while American institutions have funded 3 papers within the T100, according to the data obtained from WOS (see Figure 5).

Analyzing institutions in a bibliometric analysis is crucial as it allows for the identification of organizations that finance and support research in a specific field [22]. This information can be valuable for understanding resource allocation, funding trends, and collaboration among institutions in a particular area of research [23]. Moreover, it can help identify the most influential institutions in a field and those making the most significant contributions to scientific production in that area. In summary, the analysis of institutions provides an overview of the research landscape in a specific field and informs policies and decisions related to research investments.

## 5. Conclusions

The analysis of bibliometric data underscores the sustained significance and impact of research on cytotoxic drugs for the treatment of colorectal cancer. This is evident through the substantial volume of articles and high citation density observed in both the initial search and the subsequent refined results. Currently, research in this field is primarily concentrated in three key areas: pharmacogenetics, adjuvant substances, and advancements in pharmaceutical forms.

The leading countries actively contributing to this research are the United States and China, both of which have made substantial public investments in research endeavors. Institutions from these countries play a prominent role in funding, with the National Natural Science Foundation of China (NSFC) leading the way with funding for 13 papers, closely followed by prominent American institutions such as the National Institutes of Health and National Cancer Institute.

Collaboration among authors is common, and the predominant language of publication is English. High-impact journals are the preferred choice for publication, with the Annals of Oncology being particularly noteworthy in this regard.

## Figures and Tables

**Figure 1 cancers-15-04520-f001:**
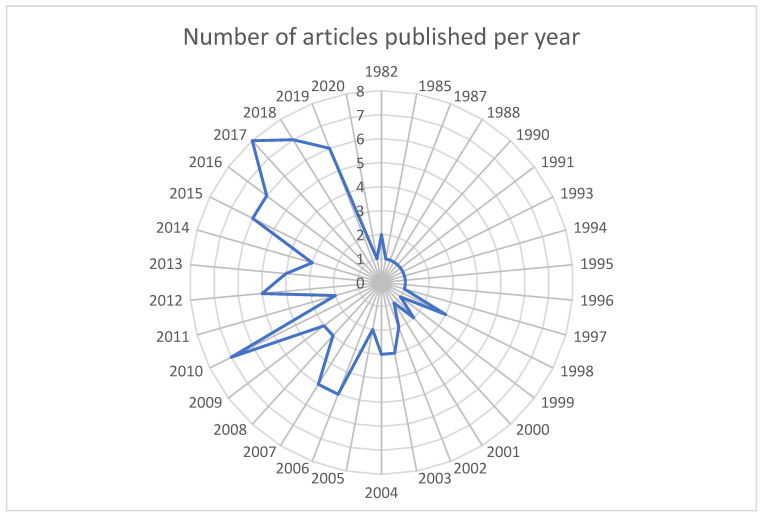
Annual distribution of T100.

**Figure 2 cancers-15-04520-f002:**
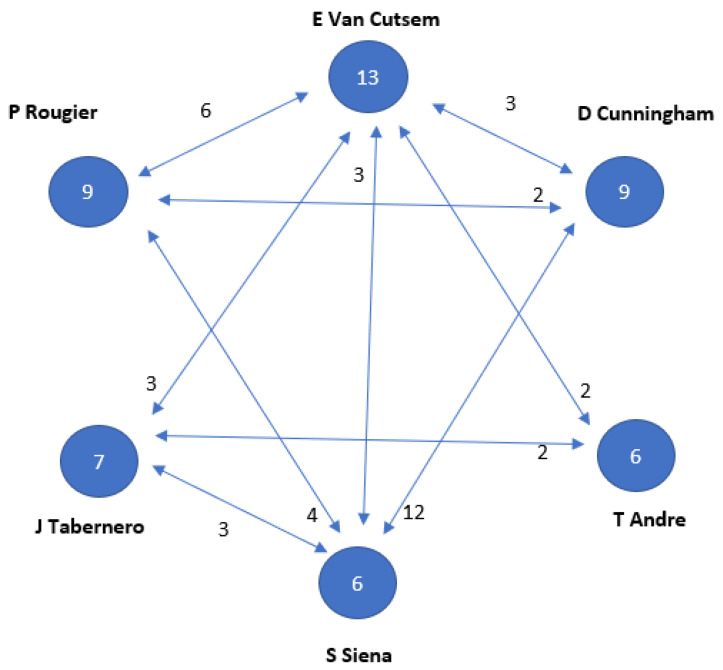
Relationship between the authors with 6 or more publications on a topic. Numbers inside the circles indicate the number of articles published. Connecting arrows and numbers affixed indicate number of papers together, respectively.

**Figure 3 cancers-15-04520-f003:**
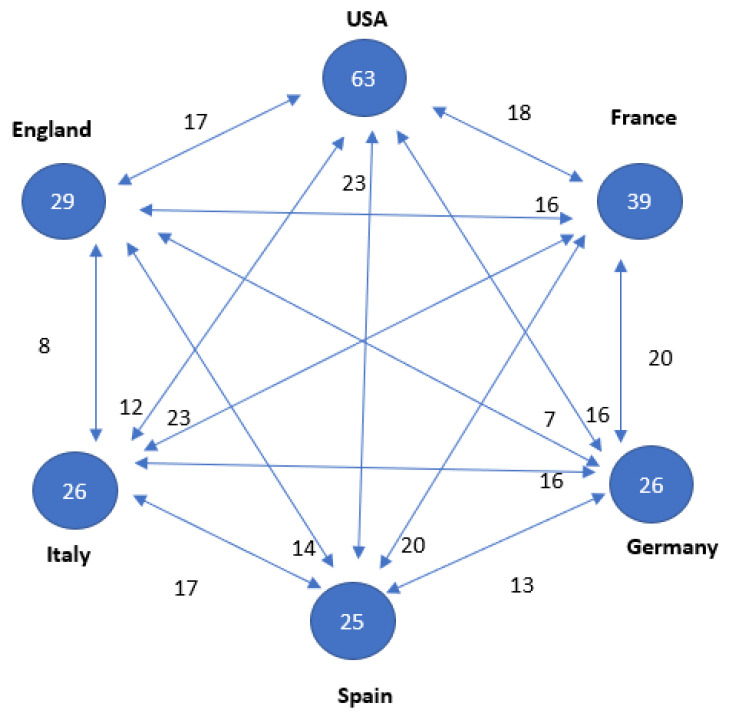
Relationship between the countries with 25 or more publications on a topic. Numbers inside the circles indicate the number of articles published. Connecting arrows and numbers affixed indicate number of papers together, respectively.

**Figure 4 cancers-15-04520-f004:**
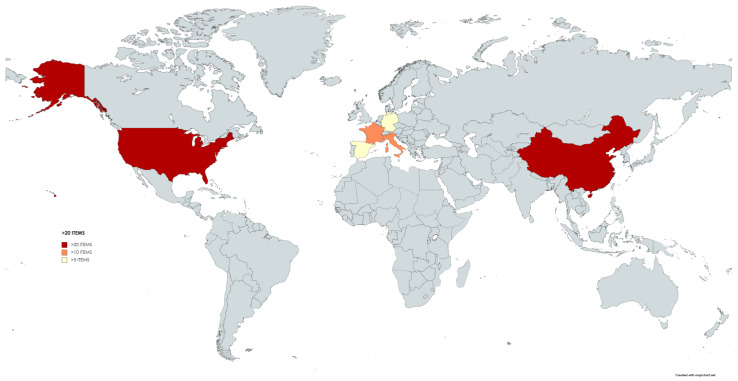
Map depicting countries with the highest number of publications within the Top 100 articles, based on a map created using MapaChart.net.

**Figure 5 cancers-15-04520-f005:**
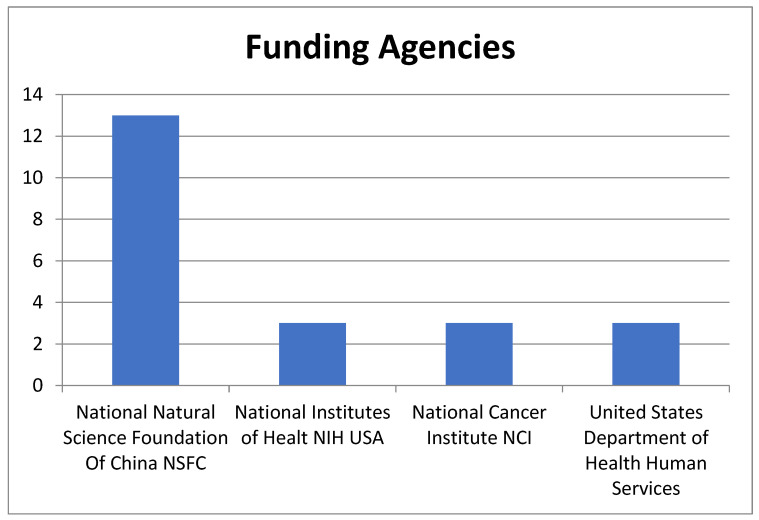
Funding agencies for more than 14 of the T100 manuscripts are arranged by the number of T100 records.

**Table 1 cancers-15-04520-t001:** Bibliometric information associated with the top 100 (T100) cited articles.

Rank	Article Title	Journal	Year	Times Quoted
#				
1	Severe hepatic sinusoidal obstruction associated with oxaliplatin-based chemotherapy in patients with metastatic colorectal cancer	Annals of Oncology	2004	778
2	Metabolism of 5-fluorocytosine to 5-fluorouracil in Human Colorectal Tumor-Cells Transduced with the Cytosine Deaminase Gene-Significant Antitumor Effects when only a Small Percentage of Tumor-Cells Express Cytosine Deaminase	Proceedings of the National Academy of Sciences of the United States of America	1994	426
3	Liver histology and surgical outcomes after preoperative chemotherapy with fluorouracil plus oxaliplatin in colorectal cancer liver metastases	Journal of Clinical Oncology	2006	410
4	Effect of steatohepatitis associated with irinotecan or oxaliplatin pretreatment on resectability of hepatic colorectal metastases	Journal of the American College of Surgeons	2005	361
5	Enhanced antitumor activity of anti-epidermal growth factor receptor monoclonal antibody IMC-C225 in combination with irinotecan (CPT-11) against human colorectal tumor xenografts	Clinical Cancer Research	2002	345
6	Liposomal curcumin with and without oxaliplatin: effects on cell growth, apoptosis, and angiogenesis in colorectal cancer	Molecular Cancer Therapeutics	2007	273
7	Chloroquine potentiates the anti-cancer effect of 5-fluorouracil on colon cancer cells	BMC Cancer	2010	265
8	Neoadjuvant treatment of unresectable liver disease with irinotecan and 5-fluorouracil plus folinic acid in colorectal cancer patients	Annals of Oncology	2004	256
9	Inhibition of autophagy augments 5-fluorouracil chemotherapy in human colon cancer in vitro and in vivo model	European Journal of Cancer	2010	255
10	Preliminary results of preoperative 5-fluorouracil, low-dose leucovorin, and concurrent radiation therapy for clinically resectable T3 rectal cancer	Diseases of the Colon & Rectum	1997	223
11	Hepatic Resection of Colorectal Metastases-Influence of Clinical Factors and Adjuvant Intraperitoneal 5-fluorouracil via Tenckhoff Catheter on Survival	Annals of Surgery	1985	221
12	Quantitative Cell-Free DNA, KRAS, and BRAF Mutations in Plasma from Patients with Metastatic Colorectal Cancer during Treatment with Cetuximab and Irinotecan	Clinical Cancer Research	2012	215
13	Interaction of Gamma-Interferon and 5-fluorouracil in the h630 Human Colon-Carcinoma Cell-Line	Cancer Research	1990	204
14	Treatment of Advanced Colorectal and Gastric Adenocarcinomas with 5-fu Combined with High-Dose Folinic Acid-a Pilot-Study	Cancer Treatment Reports	1982	190
15	MALAT1 Is Associated with Poor Response to Oxaliplatin-Based Chemotherapy in Colorectal Cancer Patients and Promotes Chemoresistance through EZH2	Molecular Cancer Therapeutics	2017	189
16	Chronicle: results of a randomised phase III trial in locally advanced rectal cancer after neoadjuvant chemoradiation randomising postoperative adjuvant capecitabine plus oxaliplatin (XELOX) versus control	Annals of Oncology	2014	184
17	Dual antitumor effects of 5-fluorouracil on the cell cycle in colorectal carcinoma cells: A novel target mechanism concept for pharmacokinetic modulating chemotherapy	Cancer Research	2001	181
18	Thymidylate Synthase Gene Amplification in Human Colon-Cancer Cell-Lines Resistant to 5-fluorouracil	Biochemical Pharmacology	1995	176
19	Digital karyotyping identifies thymidylate synthase amplification as a mechanism of resistance to 5-fluorouracil in metastatic colorectal cancer patients	Proceedings of the National Academy of Sciences of The United States of America	2004	164
20	Regulation of Thymidylate Synthase in Human Colon Cancer-Cells Treated with 5-fluorouracil and Interferon-Gamma	Molecular Pharmacology	1993	156
21	Linc00152 Functions as a Competing Endogenous RNA to Confer Oxaliplatin Resistance and Holds Prognostic Values in Colon Cancer	Molecular Therapy	2016	151
22	Melatonin synergizes the chemotherapeutic effect of 5-fluorouracil in colon cancer by suppressing PI3K/AKT and NF-B/iNOS signaling pathways	Journal of Pineal Research	2017	150
23	Prognosis of stage II and III colon cancer treated with adjuvant 5-fluorouracil or FOLFIRI in relation to microsatellite status: results of the PETACC-3 trial	Annals of Oncology	2015	149
24	Cellular determinants of oxaliplatin sensitivity in colon cancer cell lines	European Journal of Cancer	2003	145
25	Mucinous histology predicts for poor response rate and overall survival of patients with colorectal cancer and treated with first-line oxaliplatin- and/or irinotecan-based chemotherapy	British Journal of Cancer	2009	137
26	Oxaliplatin-induced peripheral neuropathy’s effects on health-related quality of life of colorectal cancer survivors	Supportive Care in Cancer	2013	134
27	Demonstration of hepatic steatosis by computerized tomography in patients receiving 5-fluorouracil-based therapy for advanced colorectal cancer	British Journal of Cancer	1998	129
28	High-Dose Folinic Acid and 5-fluorouracil Bolus and Continuous Infusion in Advanced Colorectal-Cancer	European Journal of Cancer & Clinical Oncology	1988	129
29	Capecitabine versus 5-fluorouracil/folinic acid as adjuvant therapy for stage III colon cancer: final results from the X-ACT trial with analysis by age and preliminary evidence of a pharmacodynamic marker of efficacy	Annals of Oncology	2012	125
30	Violacein synergistically increases 5-fluorouracil cytotoxicity, induces apoptosis and inhibits Akt-mediated signal transduction in human colorectal cancer cells	Carcinogenesis	2006	122
31	Prognostic Effect of BRAF and KRAS Mutations in Patients With Stage III Colon Cancer Treated With Leucovorin, Fluorouracil, and Oxaliplatin With or Without Cetuximab A Post Hoc Analysis of the PETACC-8 Trial	JAMA Oncology	2016	118
32	Association of Cell Lethality with Incorporation of 5-fluorouracil and 5-fluorouridine into Nuclear-RNA in Human-Colon Carcinoma-Cells in Culture	Molecular Pharmacology	1982	113
33	Nuclear factor-kB tumor expression predicts response and survival in irinotecan-refractory metastatic colorectal cancer treated with cetuximab-irinotecan therapy	Journal of Clinical Oncology	2007	111
34	Synergistic inhibitory effects of curcumin and 5-fluorouracil on the growth of the human colon cancer cell line HT-29	Chemotherapy	2006	108
35	dUTP nucleotidohydrolase isoform expression in normal and neoplastic tissues: Association with survival and response to 5-fluorouracil in colorectal cancer	Cancer Research	2000	107
36	Prolonged Survival of Initially Unresectable Hepatic Colorectal Cancer Patients Treated With Hepatic Arterial Infusion of Oxaliplatin Followed by Radical Surgery of Metastases	Annals of Surgery	2010	104
37	Fusobacterium nucleatum promotes chemoresistance to 5-fluorouracil by upregulation of BIRC3 expression in colorectal cancer	Journal of Experimental & Clinical Cancer Research	2019	103
38	Epigallocatechin-3-gallate targets cancer stem-like cells and enhances 5-fluorouracil chemosensitivity in colorectal cancer	Oncotarget	2016	102
39	Combination chemotherapy with combretastatin A-4 phosphate and 5-fluorouracil in an experimental murine colon adenocarcinoma	Anticancer Research	2000	101
40	Trans-arterial chemoembolization (TACE) of liver metastases from colorectal cancer using irinotecan-eluting beads: Preliminary results	Anticancer Research	2006	100
41	Study on colon-specific pectin/ethylcellulose film-coated 5-fluorouracil pellets in rats	International Journal of Pharmaceutics	2008	97
42	Outcome of posthepatectomy-missing colorectal liver metastases after complete response to chemotherapy: Impact of adjuvant intra-arterial hepatic oxaliplatin	Annals of Surgical Oncology	2007	95
43	Resveratrol induces chemosensitization to 5-fluorouracil through up-regulation of intercellular junctions, Epithelial-to-mesenchymal transition and apoptosis in colorectal cancer	Biochemical Pharmacology	2015	94
44	Evaluation of short-course radiotherapy followed by neoadjuvant bevacizumab, capecitabine, and oxaliplatin and subsequent radical surgical treatment in primary stage IV rectal cancer	Annals of Oncology	2013	91
45	Role of primary miRNA polymorphic variants in metastatic colon cancer patients treated with 5-fluorouracil and irinotecan	Pharmacogenomics Journal	2011	89
46	Sequence-dependent growth inhibition and DNA damage formation by the irinotecan-5-fluorouracil combination in human colon carcinoma cell lines	European Journal of Cancer	1999	89
47	Eudragit S100 Coated Citrus Pectin Nanoparticles for Colon Targeting of 5-Fluorouracil	Materials	2015	88
48	Adaptation to 5-fluorouracil of the Heterogeneous Human Colon-Tumor Cell-Line ht-29 Results in the Selection of Cells Committed to Differentiation	International Journal of Cancer	1991	88
49	The influence of gut microbiota dysbiosis to the efficacy of 5-Fluorouracil treatment on colorectal cancer	Biomedicine & Pharmacotherapy	2018	85
50	Curcumin mediates oxaliplatin-acquired resistance reversion in colorectal cancer cell lines through modulation of CXC-Chemokine/NF-kappa B signalling pathway	Scientific Reports	2016	85
51	High expression of microRNA-625-3p is associated with poor response to first-line oxaliplatin based treatment of metastatic colorectal cancer	Molecular Oncology	2013	85
52	Inhibition of NF-kappa B Signaling by Quinacrine Is Cytotoxic to Human Colon Carcinoma Cell Lines and Is Synergistic in Combination with Tumor Necrosis Factor-related Apoptosis-inducing Ligand (TRAIL) or Oxaliplatin	Journal of Biological Chemistry	2010	85
53	Polymorphism in the thymidylate synthase promoter enhancer region is not an efficacious marker for tumor sensitivity to 5-fluorouracil-based oral adjuvant chemotherapy in colorectal cancer	Clinical Cancer Research	2003	85
54	lncRNA KCNQ1OT1 enhances the chemoresistance of oxaliplatin in colon cancer by targeting the miR-34a/ATG4B pathway	Oncotargets and Therapy	2019	84
55	Formulation and characterization of 5-Fluorouracil enteric coated nanoparticles for sustained and localized release in treating colorectal cancer	Saudi Pharmaceutical Journal	2015	84
56	Curcumin Chemosensitizes 5-Fluorouracil Resistant MMR-Deficient Human Colon Cancer Cells in High Density Cultures	PloS ONE	2014	84
57	miR-34a mediates oxaliplatin resistance of colorectal cancer cells by inhibiting macroautophagy via transforming growth factor-beta/Smad4 pathway	World Journal of gastroenterology	2017	82
58	Oxaliplatin resistance in colorectal cancer cells is mediated via activation of ABCG2 to alleviate ER stress induced apoptosis	Journal of Cellular Physiology	2018	81
59	Induction chemotherapy with capecitabine and oxaliplatin followed by chemoradiotherapy before total mesorectal excision in patients with locally advanced rectal cancer	Annals of Oncology	2012	79
60	Abituzumab combined with cetuximab plus irinotecan versus cetuximab plus irinotecan alone for patients with KRAS wild-type metastatic colorectal cancer: the randomised phase I/II POSEIDON trial	Annals of Oncology	2015	78
61	Leucovorin Plus 5-fluorouracil-An Effective Treatment for Metastatic Colon Cancer	Journal of Clinical Oncology	1987	78
62	Preoperative treatment of patients with locally advanced unresectable rectal adenocarcinoma utilizing continuous chronobiologically shaped 5-fluorouracil infusion and radiation therapy	Cancer	1996	77
63	Treatment of colon and breast carcinoma cells with 5-fluorouracil enhances expression of carcinoembryonic antigen and susceptibility to HLA-A(*)02.01 restricted, CEA-peptide-specific cytotoxic T cells in vitro	International Journal of Cancer	2003	76
64	Biweekly cetuximab and irinotecan as third-line therapy in patients with advanced colorectal cancer after failure to irinotecan, oxaliplatin and 5-fluorouracil	Annals of Oncology	2008	74
65	Preoperative radiation with concurrent 5-fluorouracil continuous infusion for locally advanced unresectable rectal cancer	International Journal of Radiation Oncology Biology Physics	1998	74
66	Long non-coding RNA LINC00152 promotes cell proliferation, metastasis, and confers 5-FU resistance in colorectal cancer by inhibiting miR-139-5p	Oncogenesis	2017	73
67	miR-139-5p sensitizes colorectal cancer cells to 5-fluorouracil by targeting NOTCH-1	Pathology Research and Practice	2016	73
68	Inhibition of lactate dehydrogenase A by microRNA-34a resensitizes colon cancer cells to 5-fluorouracil	Molecular Medicine Reports	2015	73
69	Combination of cyclooxygenase-2 inhibitors and oxaliplatin increases the growth inhibition and death in human colon cancer cells	Biochemical Pharmacology	2005	73
70	Lactobacillus casei Variety rhamnosus Probiotic Preventively Attenuates 5-Fluorouracil/Oxaliplatin-Induced Intestinal Injury in a Syngeneic Colorectal Cancer Model	Frontiers in Microbiology	2018	72
71	miR-19b-3p promotes colon cancer proliferation and oxaliplatin-based chemoresistance by targeting SMAD4: validation by bioinformatics and experimental analyses	Journal of Experimental & Clinical Cancer Research	2017	72
72	Long noncoding RNA CRNDE functions as a competing endogenous RNA to promote metastasis and oxaliplatin resistance by sponging miR-136 in colorectal cancer	Oncotargets and Therapy	2017	70
73	Overexpression of MicroRNA-122 Re-sensitizes 5-FU-Resistant Colon Cancer Cells to 5-FU Through the Inhibition of PKM2 In Vitro and In Vivo	Cell Biochemistry and Biophysics	2014	70
74	Colon Targeting of 5-Fluorouracil Using Polyethylene Glycol Cross-linked Chitosan Microspheres Enteric Coated with Cellulose Acetate Phthalate	Industrial & Engineering Chemistry Research	2011	70
75	circHIPK3 promotes oxaliplatin-resistance in colorectal cancer through autophagy by sponging miR-637	Ebiomedicine	2019	69
76	Vitamin D mediates its action in human colon carcinoma cells in a calcium-sensing receptor-dependent manner: downregulates malignant cell behavior and the expression of thymidylate synthase and survivin and promotes cellular sensitivity to 5-FU	International Journal of Cancer	2010	69
77	Prevention of oxaliplatin-induced peripheral sensory neuropathy by carbamazepine in patients with advanced colorectal cancer.	Clinical Colorectal Cancer	2002	69
78	Insulin-like growth factor 1 expression correlates with clinical outcome in K-RAS wild type colorectal cancer patients treated with cetuximab and irinotecan	International Journal of Cancer	2010	68
79	Supramolecular Chemotherapy: Carboxylated Pillar[6]arene for Decreasing Cytotoxicity of Oxaliplatin to Normal Cells and Improving Its Anticancer Bioactivity Against Colorectal Cancer	ACS Applied Materials & Interfaces	2018	66
80	Elevated Neutrophil to Lymphocyte Ratio Predicts Poor Prognosis in Advanced Colorectal Cancer Patients Receiving Oxaliplatin-Based Chemotherapy	Oncology	2012	66
81	5-Fluorouracil and oxaliplatin modify the expression profiles of microRNAs in human colon cancer cells in vitro	Oncology Reports	2010	66
82	Epidermal Growth Factor Receptor (EGFR) gene copy number (GCN) correlates with clinical activity of irinotecan-cetuximab in K-RAS wild-type colorectal cancer: a fluorescence in situ (FISH) and chromogenic in situ hybridization (CISH) analysis	BMC Cancer	2009	66
83	The histone deacetylase inhibitor PXD101 synergises with 5-fluorouracil to inhibit colon cancer cell growth in vitro and in vivo	Cancer Chemotherapy and Pharmacology	2007	66
84	Repression of cell cycle-related proteins by oxaliplatin but not cisplatin in human colon cancer cells	Molecular Cancer Therapeutics	2006	66
85	The impact of 5-fluorouracil and intraoperative electron beam radiation therapy on the outcome of patients with locally advanced primary rectal and rectosigmoid cancer	Annals of Surgery	1998	66
86	(-)-Epigallocatechin Gallate (EGCG) Enhances the Sensitivity of Colorectal Cancer Cells to 5-FU by Inhibiting GRP78/NF-kappa B/miR-155-5p/MDR1 Pathway	Journal of Agricultural and Food Chemistry	2019	65
87	NIL_0002 Microarray Analysis of Circular RNA Expression Profile Associated with 5-Fluorouracil-Based Chemoradiation Resistance in Colorectal Cancer Cells	Biomed Research International	2017	65
88	Cannabinoid receptor-independent cytotoxic effects of cannabinoids in human colorectal carcinoma cells: synergism with 5-fluorouracil	Cancer Chemotherapy and Pharmacology	2009	65
89	miR-506 enhances the sensitivity of human colorectal cancer cells to oxaliplatin by suppressing MDR1/P-gp expression	Cell Proliferation	2017	64
90	Panitumumab combined with irinotecan for patients with KRAS wild-type metastatic colorectal cancer refractory to standard chemotherapy: a GERCOR efficacy, tolerance, and translational molecular study	Annals of Oncology	2013	64
91	Peripheral neurotoxicity of oxaliplatin in combination with 5-fluorouracil (FOLFOX) or capecitabine (XELOX): a prospective evaluation of 150 colorectal cancer patients	Annals of Oncology	2012	64
92	Dual-layered pH-sensitive alginate/chitosan/kappa-carrageenan microbeads for colon-targeted release of 5-fluorouracil	International Journal of Biological Macromolecules	2019	63
93	A novel 5-Fluorouracil targeted delivery to colon cancer using folic acid conjugated liposomes	Biomedicine & Pharmacotherapy	2018	63
94	Resveratrol Chemosensitizes TNF-beta-Induced Survival of 5-FU-Treated Colorectal Cancer Cells	Nutrients	2018	63
95	Predictive value of Ki67 and p53 in locally advanced rectal cancer: Correlation with thymidylate synthase and histopathological tumor regression after neoadjuvant 5-FU-based chemoradiotherapy	World Journal of Gastroenterology	2008	63
96	Melatonin and 5-fluorouracil co-suppress colon cancer stem cells by regulating cellular prion protein-Oct4 axis	Journal of Pineal Research	2018	62
97	Fecal Microbiota Transplantation Prevents Intestinal Injury, Upregulation of Toll-Like Receptors, and 5-Fluorouracil/Oxaliplatin-Induced Toxicity in Colorectal Cancer	International Journal of Molecular Sciences	2020	61
98	pH-sensitive ZnO/carboxymethyl cellulose/chitosan bio-nanocomposite beads for colon-specific release of 5-fluorouracil	International Journal of Biological Macromolecules	2019	61
99	Blocking heat shock protein-90 inhibits the invasive properties and hepatic growth of human colon cancer cells and improves the efficacy of oxaliplatin in p53-deficient colon cancer tumors in vivo	Molecular Cancer Therapeutics	2007	61
100	Andrographolide reversed 5-FU resistance in human colorectal cancer by elevating BAX expression	Biochemical Pharmacology	2016	60

**Table 2 cancers-15-04520-t002:** Countries publishing more than 25 of the T100 cited articles.

Country	Record Count	Times Cited	Average Citations/Record	Oldest Article
USA	25	3926	157.04	1982
China	24	2057	85.71	2006
France	12	2306	192.17	1982
Italy	11	1248	113.45	2001
Germany	7	561	80.14	2007
Spain	6	559	93.17	2011

**Table 3 cancers-15-04520-t003:** Journals publishing 4 or more than 4 of the top 100 (T100) papers are arranged by the number of T100 records.

Journal	Items	Impact Factor 2021
Annals of Oncology	11	32.97
Biochemical Pharmacology	4	5.81
International Journal of Cancer	4	7.31
Molecular Cancer Therapeutics	4	6.23

## Data Availability

The data presented in this study are available in this article.

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
