# Peer review of "Time-Tested Strategies in Modern Context: A Bibliometric Study of Chemotherapy’s Continued Importance in Colorectal Cancer Treatment"

_cancers, 2023, doi:10.3390/cancers15184520_

Round 1

Reviewer 1 Report

This is an interesting study. I am curious if you can validate your findings by doing the same analysis using the papers that ranked 101-200 based on the number of citations. In other words, I would like to see how similar (or dissimilar) your findings will be if you repeat all analyses but instead of using the first 100 articles with the highest number of citations, you use instead the next 100 articles based on number of citations.

Light editing is needed.

Author Response

We find your proposal very interesting. We conducted searches in Web of Science (WoS) and initially evaluated the data obtained from the 42,000 papers. Following your comment, we performed searches with the top 1,000 most cited papers, 500, and as you mentioned, from ranks 101 to 200 in terms of citations.

Regarding institutions or countries with the highest impact, it appears that they do not vary significantly, as the United States and China continue to be the primary contributors in each group, along with European countries. It is indeed challenging to pinpoint specific themes, especially with the 42,000 articles, where this becomes practically impossible. WoS offers the option to categorize citations as "Micro" and "Meso," but these refer to more general concepts.

Ultimately, the goal of selecting the top 100 most impactful articles is to seek a representative formula for the entirety of the research, making this type of analysis quite valuable as a starting point.

We hope you find the changes made to your liking, and we sincerely thank you for your dedication and thorough review, which enhances the quality of the work

Reviewer 2 Report

This manuscript of bibliometric study was interesting, which was associated with the topic of colorectal cancer therapy. The reviewer believed that this manuscript fell within the scope of Cancers, but the current quality was unsatisfactory. It could be further considered after a Major Revision. Please refer to the detailed comments:

Q1: It seemed that the template of Cancers was not properly used. Some formatting issues existed.

Q2: It was suggested not to divided so many paragraphs of Abstract.

Q3: There were many typos throughout the manuscript, like Line 82 “Web od Science”.

Q4: How did the authors ensure that the chemotherapy drug names had been included for literature survey as many as possible?

Q5: The format of contents in Tables should be unified.

Q6: The figure of funding agencies did not include in figure count. It should be Figure 1.

Q7: The annual distribution of articles might be more suitable to be displayed in the manner of columns.

Q8: Could the authors perform co-citation, bibliographic coupling and keywords co-occurrence analysis? Some at least preliminary results could be added.

Q9: The future research directions in this field should be discussed in Section 4.

Q10: Seen from the length of Reference list, the authors might not have been fully investigated the literature. Please consider to cite more relevant papers in this field.

There were many typos throughout the manuscript; please correct them.

Author Response

This manuscript of bibliometric study was interesting, which was associated with the topic of colorectal cancer therapy. The reviewer believed that this manuscript fell within the scope of Cancers, but the current quality was unsatisfactory. It could be further considered after a Major Revision. Please refer to the detailed comments:

I would like to extend my sincere gratitude to the reviewer for their meticulous and insightful review of my article. Their valuable feedback and constructive suggestions have significantly enhanced the quality and clarity of the manuscript. I am truly appreciative of their time and expertise, which have undoubtedly contributed to the overall improvement of this work.

Q1: It seemed that the template of Cancers was not properly used. Some formatting issues existed.

You are correct; we downloaded the original template from the Cancers journal to tailor our article to its requirements

Q2: It was suggested not to divided so many paragraphs of Abstract.

Thank you very much. We also believe it is an area for improvement. We have reworked the abstract to make it more concise and succinct while comprehensively capturing the essential content of the paper.

Q3: There were many typos throughout the manuscript, like Line 82 “Web od Science”.

We have conducted a comprehensive review to address these issues

Q4: How did the authors ensure that the chemotherapy drug names had been included for literature survey as many as possible?

For the selection of the medications subsequently analyzed in the bibliometric study, we used the cancer.gov website (An official website of the United States government) as a reference, from which we extracted both the trade name and the FDA-accepted active ingredient name.

To address the uncertainty, we have modified the paragraph in the Materials and Methods section that mentioned it

Q5: The format of contents in Tables should be unified.

Thank you very much, here is another clear point for improvement. We have reviewed, aligning the format of all the tables.

Q6: The figure of funding agencies did not include in figure count. It should be Figure 1.

You are right, we have restructured the location of different figures including the one you mention, identifying it with the corresponding number

Q7: The annual distribution of articles might be more suitable to be displayed in the manner of columns.

We think that it is a graphic license, with a purely aesthetic objective, so if it is considered that it should be changed we would agree and proceed to change it

Q8: Could the authors perform co-citation, bibliographic coupling and keywords co-occurrence analysis? Some at least preliminary results could be added.

Q9: The future research directions in this field should be discussed in Section 4.

Without a doubt it is a point that will improve the article and that therefore we are going to use, thank you very much

Q10: Seen from the length of Reference list, the authors might not have been fully investigated the literature. Please consider to cite more relevant papers in this field.

 This point is complex “per se”, due to the intrinsic methodology of what constitutes a bibliometric study. The most relevant articles are those that would appear in the bibliometric study, as their selection has been based on your citations, understanding these as a parameter of article quality and interest. When we mention them, we use symbols such as #1, #2, or #98 to refer to the article in Table T100 (top 100 most relevant articles).

We do understand the concern. Whenever we conduct a bibliometric study and review hundreds of articles, we often observe that the bibliographies are quite limited given the extent of the work. However, these articles remain referenced with the aforementioned symbols.

Round 2

Reviewer 1 Report

The authors revised appropriately.

Reviewer 2 Report

Thanks for your revision.